# Safety Leadership, Safety Attitudes, Safety Knowledge and Motivation toward Safety-Related Behaviors in Electrical Substation Construction Projects

**DOI:** 10.3390/ijerph18084196

**Published:** 2021-04-15

**Authors:** Abdulrahman M. Basahel

**Affiliations:** Department of Industrial Engineering, Faculty of Engineering, King Abdulaziz University, Jeddah 21589, Saudi Arabia; ambasahel@kau.edu.sa; Tel.: +966-12-6400-000

**Keywords:** safety leadership, safety attitudes, safety motivation, safety knowledge, safety-related work behavior, electrical construction projects

## Abstract

Poor safety conditions and performance are consequences of individual factors as well as organizational and group factors. However, little attention has been afforded to the sequential impact of these factors on safety-related behaviors (compliance and participation) in the Saudi Arabian electrical construction industry. This study examines the causal effects of leadership and attitudes on safety compliance and participation mediated by motivation and knowledge. The research collected 636 surveys in electrical construction projects for nine large contractors between November 2018 and July 2019 in Saudi Arabia. Structural equation modeling (SEM) was used to determine the mechanism by which leadership and attitudes affected safety compliance and participation through motivation and knowledge. The results indicate that safety leadership and attitude factors as well as their interactions predicted safety motivation and knowledge. Additionally, these factors affected safety participation and compliance via workers’ motivation and knowledge. Safety motivation and safety knowledge positively affected workers’ participation and compliance. Management should encourage and regularly assess effective leadership and attitudes and developing motivation and knowledge among employees can improve organizations’ safety-related behavior performance.

## 1. Introduction

A number of factors can affect safety performance in any type of organization. However, as of 2018, the number of occupational accidents due to safety matters was still high in Saudi Arabia [1]. The majority of these accidents were related to construction industry sectors and involved falling from heights (28.55%), collision with moving/stationary objects (25.21%), abrasive and friction issues (17.93%) and others (e.g., stress, overload, heat and fatigue; 28.31%) [1]. According to General Organization for Social Insurance (GOSI) [1], the total cost of medical care due to these accidents was more than $175 million. From the perspective of organizations, safety terms are almost universally left undefined, and a number of factors can lead to workplace accidents [2]. Safety at work is an issue that involves organizational/group factors (e.g., safety culture, policies, leadership and job characteristics) and individual factors (e.g., safety attitudes, knowledge, skills) [3,4]. Therefore, workplace safety is considered an attribute of the work system within organizations that is related to personal accidents and property and environmental damage [2]. A high level of improvement in construction safety has been achieved in recent decades, but accidents and serious injuries still occur among construction workers, particularly those working in electrical construction projects [5]. These types of projects involve extremely high levels of safety risks, particularly for workers who perform maintenance and construction of electrical transmission and distribution (T and D) lines, which present extremely high likelihoods of electrocution risk [5]. According to the Electrical Safety Foundation International [6], contact with overhead power lines, wiring, transformers, and contact with the electrical current of machines, tools, and appliances at construction sites can lead to serious occupational accidents such as electrocutions in electrical construction and maintenance projects. The root causes of most workplace accidents, such as contact and exposure with electrical machines and power tools at electrical construction projects, involve inadequate safety regulations, poor safety supervision, ineffective training and poor safety attitudes [7,8]. Numerous studies have considered the effect of organizational-level safety factors (e.g., safety commitment) and individual-level safety factors (e.g., safety behaviors) in various parts of the world, but in Saudi Arabia, safety issues have yet to receive much attention [9,10]. In this situation, projects are necessary to identify the proximal factors that can lead to deteriorated safety performance and increased accidents [11]. This study assessed the sequential effect of organizational factors (i.e., safety leadership; SL) and individual factors (i.e., safety attitudes; SA) on individual safety knowledge (SK) and safety motivation (SM) as precedents of safety-related behaviors (SRB).

Researchers of workplace safety have created and tested a number of workplace safety models to clarify and determine the factors that affect workplace safety [2,12,13]. These models will help to develop knowledge in the safety literature. Beus et al. [2] created a model of workplace safety that combined all of the previous theoretical reviews by safety researchers, which they called the integrative safety model (ISM). The ISM classified the aspects that influence workplace safety into two main groups of factors: organizational-/group-level factors and individual-level factors. Organizational and group factors involve contextual aspects (e.g., safety management practices, policies and safety cultures) and job characteristics (e.g., level risks and hazards, supervision and coworkers). In contrast, individual factors comprise safety attitudes, safety abilities and personal behavior. The integrated model has four segments in terms of dependency and precedence. These segments first involve distal antecedents at both the organizational/group level (e.g., safety culture/climate and job demand and hazards). Second, proximal antecedents cover organizational safety behavior, individual behavior, safety knowledge, skills and motivation. Third, leading indicators for workplace safety include safety-related work behavior (safe or unsafe). The final segment is the lagging indicator accident rate and accidents. Vinodkumar and Bhasi [14] stated that safety knowledge and safety motivation are important personal factors that influence safety behavior, in addition to safety indicators such as the number of accidents. However, workers with risky attitudes (ignoring safety rules) completing tasks in a workplace without safety incidents does not indicate safety. According to the ISM, workplace accidents depend on multiple complicated factors, such as management and group-level factors (e.g., safety culture and policy and safety supervision) [14,15] and individual-level factors (safety knowledge, safety skills, personal resources and motivation) [2]. Factors such as individual personality can affect safety performance and accidents through an individual’s levels of motivation and participation [3]. Workplace safety provides a low probability level of harm or damage to individuals, property and environments as a result of work system attributes. The majority of previous studies have considered workplace accidents as an indicator of workplace safety [2,6,11]. Indeed, workplace accidents reflect a lack of safety, but the absence of accidents does not necessarily imply workplace safety. Accidents can occur as a result of a multitude of factors (e.g., the level of safety participation, unsafe behavior and organizational aspects) [12,16]. For instance, differences in individuals’ abilities can affect motivation and safety knowledge, which can lead to poor safety behaviors and incidents [17]. It depends on a number of factors such as compliance with safety rules and regulations, participation in safety meetings and training and reporting near misses in a workplace [18]. Therefore, safety-related behavior is a proactive method that decreases future accidents in a workplace. Unsafe work behavior can be conscious or unintended; in both cases, it reflects the absence of safety conditions. In contrast, workplace accidents reflect the absence of safety after damage/injury occurs. Therefore, safety-related work behaviors are more proximal indicators than accidents since they are considered preceding factors of accidents [3,19]. Safety-related behaviors and accidents are two indicators of work safety in construction projects, but safety-related behaviors are more informative and can help to identify a lack of safety in the workplace before injury/damage occurs [12,19]. For that reason, the present research study covers safety-related behavior factors to predict workplace safety in electrical construction projects and as a preceding factor for accidents. Safety behaviors are reflected by safety compliance and safety participation [18]. Safety compliance refers to performing tasks safely to maintain workplace safety, such as by using personal protective equipment and complying with safety rules. Safety participation includes discretionary attitudes that help to maintain workplace safety and usually is considered an indirect variable, such as attending regular safety meetings, creating safety near-miss reports and alternating coworkers for unsafe conditions.

Numerous researchers have noted that the greatest determinants of safety-related behaviors for task performance are safety knowledge, skills and motivation [20,21,22]. However, the majority of previous research studies have directly studied the relationships between individual differences (e.g., safety attitudes and personality traits) and safety-related behaviors without considering mediating variables such as safety knowledge and safety motivation [2,23]. Additionally, there is a lack of literature on the interactive effect among group-level safety factors (e.g., safety leadership) and individual-level factors (e.g., safety attitudes) in predicting individual safety-related behaviors (safety compliance and safety participation) via safety knowledge and motivation. For instance, no study has examined the impact of individual safety factors such as safety attitudes and knowledge as attenuated by safety leadership [2]. Based on previous research and a review by Beus et al. [2], no previous research has evaluated the mechanism of sequential effects of group-level safety factors (e.g., safety leaderships) and individual-level safety factors (e.g., safety attitudes) on safety-related behavior (safety compliance and safety participation) with regard to safety motivation and knowledge, particularly in the high-risk industrial sector (e.g., electrical construction projects).

A number of research studies have found that SL significantly influences safety behavior and accident rates [24,25]. The safety leadership factor is a group-level factor that can impact the individual level and safety performance [24]. For instance, effective safety leadership behavior can positively impact safety performance through discussions of safety issues with workers and by providing valuable guidance and direction for a safe workplace [22,24]. The proactive and visible safety behavior of leaders leads to improved workforce safety performance in terms of compliance with rules and regulations and participation in safety trainings and meetings [26,27]. According to Lu and Yang [21], there are two types of leadership behaviors: task-oriented (i.e., relying on the match between rewards and performance) and relationship-oriented (i.e., relying on future development). It has been stated that a high level of safety motivation by leaders has a significant effect on the safety-related behaviors of the workforce [28,29,30]. Leader activities such as proposing incentive programs, recognizing workers’ safety behaviors, considering workers’ decisions, and proposing effective safety training programs and opinions related to safety issues effectively impact individuals’ safety attitudes, motivation and knowledge [24,30]. According to Turner et al. [31], managers’ activities, such as helping workers perform their tasks safely and providing information about safety improvements in the workplace [32], can buffer the negative impact of poor individual attitudes toward safety. According to previous studies, the relationship between group-level safety factors (e.g., safety leadership) as an antecedent to workers’ safety knowledge and motivation is unclear because the majority of studies have considered the direct relationship between this factor and safety performance [2,24,33]. In this study, safety leadership is considered as a group-level factor that influences individuals’ safety knowledge and motivation significantly and affects individuals’ safety-related behaviors (compliance and participation).

Individuals’ willingness to comply with workplace safety rules and regulations and to approve of safety behavior is referred to as safety motivation. The effects of personal characteristics, attitudes and self-efficacy on safety motivation has been demonstrated, although no study has assessed the impact of poor safety attitudes on motivation, as attenuated by a positive group-level safety factor (i.e., safety leaderships) [13,34,35]. Lue and Yang [32] found that leadership was significantly related to safety motivation and significantly influenced safety participation and compliance. Relationships exist between leadership and safety-related behaviors (compliance and participation), although no relationship has been found between safety policy and safety leadership [24,36]. Safety attitudes, such as individuals’ satisfaction with environmental safety work conditions, are related positively to safety motivation, individual safety performance and accident rates [37] and safety knowledge [38]. Safety attitudes can be used as a predictor of safety performance (safety-related behaviors) and accident rates [37,39]. Individuals in the workplace may not intend to become involved in incidents; however, the behavior that leads to these incidents is intentional [40]. Generally, individual safety satisfaction in the workplace is a result of high safety attitudes. Individuals’ satisfaction with some aspects of safety such as safety rules and procedures, management and leaders’ commitment to safety issues, training and precautions leads to increased safety attitudes, which significantly affects performance and accident rates [37].

Based on the previous literature, the present study aims to examine the relationships between group-level factors, individual-level factors and safety-related behavior performance. In order to understand the sequential effects of a group-level safety factor (i.e., safety leadership) and individual-level safety factors (i.e., safety attitudes) and behaviors (safety compliance and safety participation) mediated by safety motivation and knowledge using structural equation modeling, the structural model was built in the present study as illustrated in Figure 1. In addition, the present paper explores the sequential interaction effect of safety leadership and individual safety attitudes on safety compliance and participation via motivation and knowledge. Therefore, the following hypotheses are proposed:

**Hypothesis 1** **(H1).**
*Safety motivation will be predicted by safety leadership and safety attitudes.*


**Hypothesis 2** **(H2).**
*Safety knowledge will be predicted by safety leadership and safety attitudes.*


**Hypothesis 3** **(H3).**
*Safety leadership/safety attitudes with respect to safety motivation will be positively related to safety participation.*


**Hypothesis 4** **(H4).**
*Safety leadership/safety attitude with respect to safety motivation will be positively related to safety compliance.*


**Hypothesis 5** **(H5).**
*Safety leadership/safety attitude with respect to safety knowledge will be positively related to safety participation.*


**Hypothesis 6** **(H6).**
*Safety leadership/safety attitude with respect to safety knowledge will be positively related to safety compliance.*


**Hypothesis 7** **(H7).**
*The interaction effect of safety leadership and safety attitudes will significantly positively influence safety participation and compliance with respect to safety motivation and knowledge.*


**Figure 1 ijerph-18-04196-f001:**
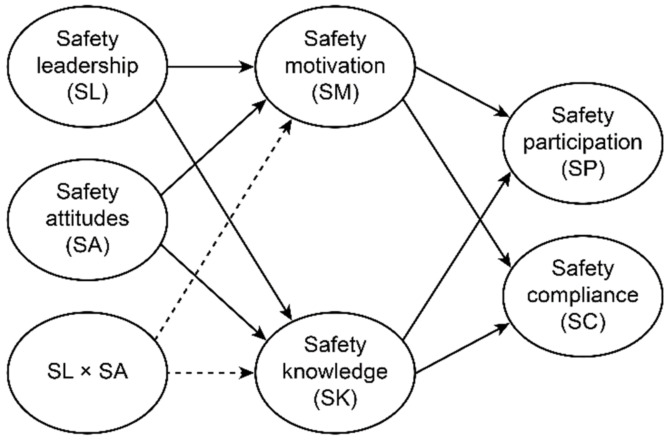
The proposed structural model (main model): the solid line represents the proposed correlations of H1 to H6, and dotted lines are modeled to examine the exploratory proposed correlations of the two-factor interactions (H7).

## 2. Materials and Methods

### 2.1. Sample Size

Data were collected from different electrical project contractors and construction projects in Saudi Arabia. The questionnaires were distributed to 920 workers in nine large electrical project contractors and were collected through email and a sealed collection box assigned to the contractors’ top management. The data were collected from November 2018 to July 2019. At some construction sites, safety advisors clarified the questionnaire for the participants. Confidentiality was ensured, and the participants could withdraw from participation any time. A follow-up was mailed, and the workers were contacted directly by top management. Three types of demographic data were observed, workers’ age, individual work experience in construction projects and level of education, as illustrated in Table 1.

The majority of participants had work experience ranging from 3 to 5 years. Additionally, most of them had a diploma degree (39.1%). A number of accidents were observed in nine electrical projects from 2016 to 2019 (see Table 1). The types of accidents were divided into three categories: minor, major and fatal. Minor accidents refer to any simple injury that requires first aid, whereas major injuries lead to amputation, loss of any body part, hospitalization of the individual within 24 h, or temporary impediment of individual work/ability of movement [41,42]. A total of 636 questionnaires were sufficiently completed and returned, which led to a 69.0% usable response rate, for a total of 636 male participants in this research. The percentage of valid participants in this research study is similar to previous research studies that applied structural equation modeling [15,24,43]. All respondents who participated in this study were engineers, safety inspectors and workers.

### 2.2. Measures

This research study aimed to investigate the sequential effects of leadership and individual attitudes on safety-related behaviors via safety motivation and knowledge. Measurements were obtained from previous research studies to ensure the reliability and validity of the study. The questionnaire involved group and individual safety measures (safety leadership and safety attitudes, respectively), individual safety motivation and knowledge, and safety-related behaviors (safety compliance and safety participation). Each output measure involved different items: 16 items assessed SL, 19 items measured SA, and 6 items measured for SM as well as for SK. Safety compliance (SC) and safety participation (SP) were measured with 5 items, as shown in Table 2. A five-point Likert scale with anchors ranging from “1 = strongly disagree” to “5 = strongly agree” was used with all measures (safety leadership, safety attitudes, safety motivation and knowledge, safety compliance and safety participation). This type of scale is commonly used in questionnaire-based academic research studies [44]. Additionally, interviews and discussion with three safety officers and four safety advisors (with more than 15 years of experience) in three electrical construction projects were conducted to support the validity of the questionnaires. Modifications to some items were considered, such as rephrasing and rewording the items to be suitable for the local working conditions and culture. The accident numbers (minor, serious and fatal accidents) were collected from the nine large construction contractors, as shown in Table 1. A large contractor was identified depending on the size of the project budget and the number of workers (i.e., 1000 or more).

To measure SL, sixteen items that covered three main aspects related to senior managers’ safety motivation, safety policy and safety concerns were adapted from Lu and Yang [24] and Wu et al. [29]. Examples of these items are “My senior manager establishes clear safety goals”, “My senior managers emphasize worksite safety”, “My senior managers stress the importance of wearing personal protective equipment”, “My senior managers have set up a safety-intensive system”, and “My senior managers trust workers”. The Cronbach’s alpha of this measure was 0.87, as shown in Table 3, which indicates a positive impact of safety leadership on individual safety motivation and safety knowledge.

Individual SA was measured by using the Safety Attitudes Questionnaire (SAQ) adapted from Donald and Canter [40]. The validity and reliability of this questionnaire have been determined in a number of research studies [37]. The Cronbach’s alpha of this main measure was 0.81, as presented in Table 2. Six subscales were considered in this study. The questionnaire comprised three items on worksite satisfaction with the safety system (e.g., “My workmates and I are satisfied with the safety procedures in general”); four items on housekeeping and safety equipment (e.g., “Before I start work, I check the safety equipment I might need”); four items on worksite encouragement and support (e.g., The people I work with encourage me to work safely”); three items on shopfloor training (e.g., “I feel satisfied with the attention given to safety in any training I have had”); two items on the level of safe working behavior (e.g., “Overall, I think I work safely”); and three items on safety information (e.g., “The people I work with are satisfied with the information they get about safe working”). The Cronbach’s alphas of all these subscales were 0.82, 0.76, 0.88, 0.90, 0.71, 0.74 and 0.83, respectively.

SM for this study used five items adapted from Vinodkumar and Bhasi [14]. Example items are “I feel that it is important to maintain safety at all times” and “I believe that safety in the workplace is a very important issue”. The Cronbach’s alpha of this measure was 0.80. SK was measured with six items adapted from Guo et al. [43] and Vinodkumar and Bhasi [14]. Example items are “I know how to perform my job in a safe manner” and “I know how to reduce the risk of accidents and incidents in the workplace”. The Cronbach’s alpha of this measure was 0.86.

SC for this study used five items adapted from Vinodkumar and Bhasi [14] and Turner et al. [31]. Example items are “I always carry out my work in a safe manner” and “I use all necessary safety equipment to do my job”. The Cronbach’s alpha of this measure was 0.92. SP was measured by five items adapted from Vinodkumar and Bhasi [14] and Turner et al. [31]. Examples of items are “I always point out to the management if any safety-related matters are noticed in my company” and “I encourage my coworkers to work safely”. The Cronbach’s alpha of this measure was 0.88.

### 2.3. Data Analysis

The structural equation modelling (SEM) technique was used to analyze the data [45]. AMOS software (version 22) (IBM, New York, NY, USA) was used to examine the causal sequence impact of item measures: safety leadership at the macroorganizational level and individual safety attitudes on safety-related behavior variables (safety compliance and safety participation) via the microlevel variables of safety motivation and safety knowledge. SEM is a useful technique to assess a series of variable dependency relationships simultaneously [46]. Moreover, SEM is conducted to compare several competing models that are proposed and to select the best-fitting and most significant one [43]. Therefore, the present study used SEM to examine the causal sequence effect of these item measures as well as the relationships. Additionally, the present study aimed to select the most parsimonious model that represented the relationships of the sequential effect of safety leaderships and individual safety attitudes on safety compliance and safety participation via safety motivation and safety knowledge. However, to assess the fit of SEM models, the present study used a number of fit indexes since there has been no consensus among researchers regarding the best index of overall fit for assessing SEM models [43,46]. These fit indexes are divided into absolute fit indexes and incremental fit indexes [24,46]. Absolute fit indexes include χ^2^, χ^2^/degrees of freedom (*df*), and root mean square error of approximate (RMSEA). These absolute indexes are used to evaluate the badness of fit for SEM models, and a value close to zero indicates an optimal model fit [47]. The χ^2^ is influenced by the sample size because it should be associated with χ^2^/degrees of freedom and the *p*-value [48]. Since the fit index (χ^2^/degrees of freedom) is less sensitive to sample size, it can address the sample size limitation [43]. The fit of the SEM model is accepted if the value of χ^2^/*df* is lower than 2 [49]. RMSEA was used in this study and is considered one of the most informative parameters in covariance structure modeling [50]. In the present study, incremental fit indexes were calculated using the comparative fit index (CFI), Tucker-Lewis index (TLI) and incremental index of fit (IFI). To make a comparison between the hypothesized model and competing models, the CFI index was used, which is the same as the normed fit index (NFI); both range from 0 to 1 [51,52]. A CFI value greater than 0.95 indicates a well-fitting model [43]. IFI and TLI index values are used to indicate the level of model goodness; the IFI index was created by Bollen [53]. IFI and TLI index values should be close to 0.95 to consider a model a good fit [54]. SPSS-10 software (IBM, NY, US) was used to determine the Cronbach’s alpha (α) values to assess the goodness of the item measures. The minimum acceptable level of α that reflects a good level of reliability is 0.6 [14,55]. In addition, the unidimensionality test was conducted to assess the strongest level of unidimensionality of all scales used in the present study. This test helps to check the availability of a single item measure underlying a set of measures [55]. A strong level of unidimensionality is examined by the CFI and should be equal to or greater than 0.9 [14,24]. SPSS-10 software was used to conduct an ANOVA to determine the effect of output measures (safety leadership, safety attitudes, safety motivation, safety knowledge, safety compliance and safety participation), age and descriptive statistics (age, work experience in years and education degree) on the type of accident.

## 3. Results

### 3.1. Reliability Analysis and Correlations among Output Measures

As illustrated in Table 2, all Cronbach’s alpha values of the item measures exceeded 0.8, which indicated the internal consistency (reliability analysis) of the proposed structural model. All the CFIs of the item measures were greater than 0.9, and one measure was equal to 0.9, as shown in Table 2. Factor loading analysis was conducted for all item measures of safety leadership, individual safety attitudes, safety motivation, safety knowledge and safety-related behaviors (safety compliance and safety participation). All the loading values that related the predictors to the latent variables were statistically significant (*p* < 0.01), as shown in Table 3.

A correlation was found between safety output measures (safety leadership, safety attitudes, safety motivation, safety knowledge, safety compliance and safety participation) and descriptive statistics (age, work experience in year, education degree and type of accident). The means, standard deviations and relationships between all item measures are illustrated in Table 4. The relationship analysis concluded that the correlation between safety leadership, motivation, and knowledge was significantly positive, as was the correlation between safety attitudes, motivation and knowledge (*p* < 0.01) (as shown in Table 4). Therefore, these results supported hypotheses H1 and H2. Additionally, safety motivation and safety knowledge were significantly positively correlated with safety compliance and safety participation (*p* < 0.01 and *p* < 0.05, respectively), which supported hypotheses H3, H4, H5 and H6_._ The participants had the highest mean score on safety compliance (4.38), followed by safety attitudes (4.28). Additionally, as shown in Table 4, the safety attitude was significantly positively correlated with age, work experience and education level (*p* < 0.01, *p* < 0.05 and *p* < 0.05, respectively). SL was positively correlated with work experience (*p* < 0.05) and education level (*p* < 0.01). SM was significantly negatively correlated with education level, whereas SK was significantly related to work experience and education level. The negative significant correlation between work experience vs. SP and work experience vs. accident numbers (*p* < 0.05).

The comparison analysis results (ANOVA) of the mean at a significance level of 0.05 between contractors showed safety leadership, safety attitude, safety knowledge, safety motivation, safety compliance and safety participation. Contractor 5 had the highest mean safety leadership, attitudes, compliance and participation (*p* < 0.05), followed by contractors 1 and 8. On the other hand, contractors 9 and 7 had the lowest mean values in safety leadership, attitudes, compliance and participation. However, contractors 2, 3, 5 and 6 had the highest mean safety motivation and knowledge (*p* < 0.05). According to the accident number analysis (minor, major and fatal) from 2016 to 2019, the lowest number of accidents was found for contractor 5 (168 accidents), as shown in Table 5, which had the highest mean safety leadership, attitudes, compliance and participation, followed by contractor 1 (226 accidents), as shown in Table 5. Contractor 9 included the highest number of accidents (433) and had the lowest safety leadership, attitude, compliance, and participation scores, followed by contractors 4 and 7 with accidents 423 and 414, respectively.

### 3.2. Structural Model Analysis

The hypothesized two-factor model was tested and involved the indirect impact of safety leadership (group-level factor), safety attitudes (individual-level factor) and the interaction of both on safety participation and safety compliance via safety motivation and safety knowledge. According to the literature review, there were various relationships between latent factors. The model posited by Lu and Yang [24] showed a positive effect of the safety leadership factor on safety compliance and participation via safety motivation and concerns. In addition, it has been highlighted that positive relationships exist between individual safety attitudes and safety motivation [17,56]. Furthermore, individual differences exist, such as attitudes and personality traits vs. safety knowledge [23]. Flin and Yule [30] and Yule et al. [57] noted that there is a significant association between effective leader supervision and safety performance (safety compliance and safety participation). Therefore, the present study assessed six alternative competing models in addition to the main model, as illustrated in Table 6. Model 1 proposed the sequential influence of safety leadership, safety attitude factors and the effect of the interaction of both factors, similar to the main model, and considered the direct influence of SL on safety compliance and safety participation. Model 2 assumed the same relationships as in model 1 among the latent factors with the addition of the direct effects of SA on safety compliance and safety participation. Model 3 posited the same correlations between factors as in model 2, adding the direct impact of SL × SA on safety compliance and safety participation. Model 4 considered the indirect impact of SL and SA and removed the impact of SL × SA. Model 5 had the same latent factor correlations as model 4 with the addition of direct effects of SL on safety compliance and safety participation. Model 6 had the same factor relationships as model 4 with the addition of the direct effects of SA on safety compliance and safety participation.

The results of the overall fit, shown in Table 6, involved the comparisons of competing models with the main model. The main model fit index values were χ^2^ = 673.45, χ^2^/*df* = 1.23, RMSEA = 0.048, CFI = 0.962, TLI = 0.952 and IFI = 0.963. Compared with model 1 (χ^2^ = 735.33), the main model was significantly different and showed a significantly better fit than model 1. Furthermore, the direct impact of safety leadership on safety compliance and safety participation (i.e., model 2; χ^2^ = 708.42) did not lead to a significant improvement in the model fit level. On the other hand, model 3, with fit index values of χ^2^ = 678.83, χ^2^/*df* = 1.28, RMSEA = 0.049, CFI = 0.958, TLI = 0.950 and IFI = 0.958, did not show significant differences in fit level from the main model, indicating that adding the direct effect of safety leadership and safety attitude interaction on safety compliance and safety participation did not result in a reduction in the model fit. According to the comparison between the main model and model 3, in terms of reasonable cost and simple relationship, the main model was better. Therefore, this study found that the main model is a final model (see Figure 2) that can represent the observed relationships between safety factors. Removing the predictive indirect effect of the leadership and attitude interaction on safety compliance and safety participation, as proposed in model 4, did not lead to a significantly higher fit model level than the main model. Similarly, the model fit differences between the main model vs. model 5 and the main model vs. model 6 were significant, with a decrease in models 5 and 6 (χ^2^ = 744.56, Δχ^2^ = 71.11, *p* < 0.001) and (χ^2^ = 731.48, Δχ^2^ = 58.03, *p* < 0.01), as presented in Table 6.

Figure 2 shows the finalized model (i.e., the superior model). Safety motivation was positively impacted by safety leadership (β = 0.61, *p* < 0.001) and individual safety attitudes (β = 0.58, *p* < 0.001). Additionally, safety leadership and safety attitudes were positively affected by safety knowledge (β = 0.47, *p* < 0.001 and β = 0.51, *p* < 0.001, respectively). These results were consistent with H1 and H2, which hypothesized that both safety motivation and safety knowledge were predicted by safety leadership and safety attitudes. The model analysis results showed that a high level of safety participation and safety compliance were associated with a high level of individual safety motivation, β = 0.36, *p* < 0.01 and β = 0.53, *p* < 0.001, respectively, which supported the hypotheses (H3 and H4). As expected, the estimated coefficient was positive between safety knowledge vs. safety participation (β = 0.43, *p* < 0.001) and safety knowledge vs. safety compliance (β = 0.55, *p* < 0.001); therefore, hypotheses H5 and H6 were validated. Finally, the hypothesized link (H7) between the safety leadership and safety attitude interaction vs. safety motivation as well as between the safety leadership and safety attitudes interaction vs. safety knowledge was positive (β = 0.48, *p* < 0.001 and β = 0.38, *p* < 0.01, respectively).

## 4. Discussion

This research study was aimed at examining the mechanism of the sequential effect of safety leadership (i.e., a group-level factor) and safety attitudes (i.e., an individual-level factor) on workers’ safety-related behaviors (safety participation and safety compliance) via safety motivation and safety knowledge in electrical construction project settings. Safety leadership was significantly and positively correlated with individual safety motivation and safety knowledge, which was validated H1 and H2. Furthermore, safety attitudes were impacted positively by individual safety motivation and safety knowledge (H1 and H2). The results indicated that high levels of safety participation and safety compliance were associated with high levels of safety leadership and safety attitudes. Previous studies have recognized the impacts of safety leadership and individual safety attitudes on safety compliance and safety participation [24,33]. In addition, the findings indicated that high levels of performance in individuals’ safety participation and safety compliance were directly related to high levels of individual safety motivation and safety knowledge. The results of the current study showed that motivation fully mediated the effects of leadership and attitudes on safety compliance and participation; thus, hypotheses H3 and H4 were supported. Safety leadership was significantly correlated with safety motivation and significantly affected safety participation and safety compliance [24]. Safety leadership and safety attitudes had a significant effect on safety related-behavior performance through enhancing safety knowledge; therefore, H5 and H6 were supported. These results were consistent with previous studies demonstrating the positive impact of management leadership on safety-related behaviors via safety motivation [24,27] and safety knowledge [2,14]. Additionally, improvements in safety attitudes had significant effects on safety compliance and safety participation through enhancing safety motivation [37] and safety knowledge [38]. Furthermore, the interaction of safety leadership and attitudes predicted safety motivation and knowledge, which validated H7. Because effective leadership contributes to a good level of supervision with proper types of training and clear safety rules and guidelines and encourages individuals to participate in safety issues, it is expected to improve safety motivation and knowledge [24,25,43]. The direct effects of leadership and attitudes on compliance and participation were not significant. This may have been because effective supervision at the group level, such as by involving workers in safety matters and encouraging workers to work safely, as well as rewards programs for safe working all positively influenced the levels of individual motivation and knowledge, which led to improved individual safety performance [14]. A reasonable explanation is that high levels of individual safety satisfaction and attitudes toward safety matters, such as the level of housekeeping in the workplace, the types of safety training and the management’s commitment to safety issues, lead to improved safety motivation and knowledge, which decreases accidents by improving safety motivation, skills and knowledge [37,39]. As expected, both safety motivation and safety knowledge are strongly and positively related to safety compliance and safety participation. Therefore, these results completely support Campbell’s [58] performance theory.

Electrical construction projects are one of the riskiest projects of the industrial sector in Saudi Arabia. The managers of these types of projects make great efforts to ensure safe working conditions; however, in general, they are not able to prevent accidents. The present study tested a number of hypotheses that explained the roles of the effects and sequential relationships of leadership and attitudes at the individual level, with compliance and participation mediated by individual safety motivation and knowledge in the electrical construction project context. According to the literature review, this is the first study to provide empirical evidence of the mechanism by which leadership and attitudes as well as their interaction affect safety compliance and participation while considering safety motivation and knowledge factors in the electrical construction sector.

### 4.1. Implication of the Study Findings

Several implications can be highlighted for organizations from the present study results. First, the results indicate the importance of effective leadership as a group-level. The findings indicate that leadership actions such as a clear safety policy and rules, a supportive work environment, rewards for safe working, encouraging workers to be involved in safety issues, regular monitoring of compliance with safety rules and instructions have a significantly positive effect on individual safety motivation and knowledge and therefore enhance safety compliance and participation among electrical construction project workers. Second, most previous research studies have demonstrated the important effect of organizational factors, such as the safety climate, on safety performance [31,59]; however, less attention has been paid to group-level factors (e.g., safety leadership) and individual-level factors (e.g., safety attitudes) as determinants of safety compliance and behavior via safety motivation and knowledge [2], which was examined in the present study. By understanding the impact of leadership as well as the importance of individual safety attitudes on workers’ safety motivation, knowledge and performance, electrical construction managers can create an effective action plan with which to enhance the levels of motivation and safety knowledge among their workers. Additionally, understanding the importance of how safety leadership and safety attitudes affect safety performance can encourage safety management to propose different parameters with which to assess these factors, regularly record the required related information (unsafe behavior, worker errors, housekeeping conditions, safety feedback and satisfaction with job safety) and perform analyses as proactive plans. In addition, the results illustrate that safety supervisors can consider the idealized (i.e., positive) influence behavior personality, developing workers’ goal achievement through inspirational appeals, having workers participate intellectually in problem solving and recognizing safety feedback and safe practices, all of which will improve safety motivation and knowledge. Third, the study findings indicate that safety motivation and knowledge were positively associated with safety-related behaviors [24,29], compliance and participation. These results suggest that better leadership leads to reduced accidents and improved safety performance and that workers having good safety attitudes improves safety performance and reduces human errors [37].

### 4.2. Limitations of the Study and Future Research

The present study was conducted in a specific period; therefore, the findings of the study reflect only a particular moment in time. As a result, future research should consider longitudinal studies to examine the effect of leadership and individual safety attitudes on safety-related behavior performance among electrical construction project workers. Second, the data collection depended on the self-reports and perceptions of the participants toward safety leadership and attitudes. Therefore, the willingness level of the workers may have affected their responses. Additionally, there was a lack of objective measures at the contractor level for specific occupational tasks, such as worker compliance percentage and participation in safety training, that could reflect workers’ actual safety performance. Future studies should consider measures that can better reflect safety behavior performance, such as the percentage of worker compliance with safety rules and procedures. Third, the present study examined leadership and individual attitudes. Future studies should consider other organizational-level factors, such as the safety culture or climate, and individual factors, such as skills and mental or physical capabilities, to predict safety behavior. Several previous research studies have proposed that a safety culture or climate can predict safety behavior performance [2,15,60] and that safety skills can affect safety behavior performance [14]. Finally, factors such as a contractor’s experience in the same type of electrical construction projects, the types of contractual agreements and the level of job satisfaction among workers can lead to significant differences in the assessments of safety leadership and safety attitudes. Therefore, future research should consider explanations of how factors such as these can influence safety performance while considering measures of safety leadership and safety attitudes.

## 5. Conclusions

The present study examined and hypothesized the complex associations and sequential influences of SL as a group-level factor and SA as an individual-level factor as well as the interaction of SA × SA on safety performance (i.e., SC and SP performance) mediated by the individual-level factors of SM and SK. This study is the first to examine the associations between these factors as part of the ISM [2] in developing regions. The majority of previous studies have considered organizational/group safety factors and individual safety factors separately [2,14]. To test and validate the hypothesized safety performance model (SC and SP) and to examine the relationships between the causal factors of SC and SP, SEM was conducted. The results provided solid empirical evidence for the theoretical model that antecedents, factors and determinants of safety performance are strongly associated. The study findings demonstrated the validity and reliability of the two different safety level factors, two determinants and two elements of safety performance. The results showed an impact of perceptions of SL and SA on safety performance via their influence on SM and SK. In addition, SM and SK were found to predict safety performance. Furthermore, the sequential interaction effect of SA × SA on safety performance through SM and SK was demonstrated. The indirect effect of two safety factors, SL and SA, and the interaction of both factors on safety performance were confirmed, although a direct impact did not exist. With regard to the type of accidents, the results suggested that accidents frequently occur in electrical contracting projects that involve low perceptions of SL and SA on safety performance. For example, contractor 9 was observed to have the highest number of accidents during the 2016–2019 period; in contrast, the lowest number of accidents was found for contractor 5, which showed the highest SL and SA scores. This study highlighted the importance of effective safety supervision together with promoting positive safety attitudes among electrical construction project employees to improve the safety of the workplace.

## Figures and Tables

**Figure 2 ijerph-18-04196-f002:**
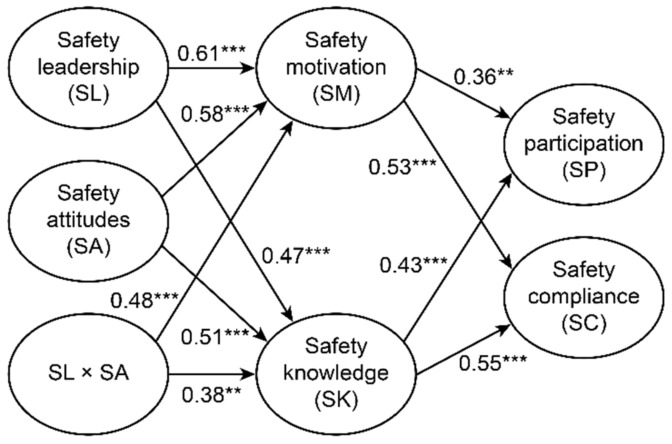
Standardized path coefficients for the finalized model (Note: for clarity, error terms and factor loadings are not presented; ** *p* < 0.01 and *** *p* < 0.001).

**Table 1 ijerph-18-04196-t001:** The demographic characteristics of the participants in the nine electrical project contracts.

Variables	Questionnaire Participants (*n* = 636)
Age, years; no. (%)	
≤20	96 (10.9)
21–30	178 (28.0)
31–40	194 (30.5)
41–50	112 (17.7)
>50	56 (8.8)
Work experience, years; no. (%)	
<3	74 (11.7)
3–5	212 (33.4)
6–10	99 (15.6)
11–15	182 (28.7)
>15	69 (10.9)
Education degree; no. (%)	
Under primary school	37 (5.8)
Primary school	92 (14.5)
High school	147 (23.1)
Diploma	248 (39.1)
Degree or higher education	112 (17.6)
Type of accident, from 2016–2019; no.	
Minor	2209
Major	688
Fatal	27

**Table 2 ijerph-18-04196-t002:** Factor loading analysis for the items underlying each output measure.

Item Measures	Factor Loading
SL: Safety Leadership	
SL1: My senior managers trust workers.	0.82
SL2: My senior managers reward those who set an example in safety behavior.	0.93
SL3: My senior managers praise workers’ safety incentive system.	0.78
SL4: My senior managers have set up a safety incentive system.	0.76
SL5: My senior managers encourage workers to report potential incidents without punishment.	0.84
SL6: My senior managers encourage workers to provide safety suggestions.	0.70
SL7: My senior managers encourage workers’ participation in safety decision-making.	0.91
SL8: My senior managers explain the safety mission clearly.	0.92
SL9: My senior managers emphasize worksite safety.	0.77
SL10: My senior managers have established a safety responsibility system.	0.83
SL11: My senior managers establish clear safety goals.	0.89
SL12: My senior managers stress the importance of wearing personal protective equipment.	0.79
SL13: My senior managers express an interest in acting on safety policies.	0.77
SL14: My senior managers are concerned about safety improvement.	0.92
SL15: My senior managers coordinate with other department to solve safety issues.	0.90
SL16: My senior managers show consideration for workers.	0.88
SA: Safety Attitudes	
SA1: My workmates are satisfied with the safety procedures in general.	0.96
SA2: I am satisfied with safety equipment in the workplace.	0.92
SA3: I am satisfied with the safety precautions that are applied in the workplace.	0.83
SA4: Before I start work, I check the safety equipment I might need	0.92
SA5: I am satisfied with the level of housekeeping in the workplace.	0.95
SA6: I am satisfied with the maintenance level of my personal protective equipment (PPE).	0.88
SA7: I return the equipment to the assigned place after use.	0.93
SA8: The people I work with encourage me to work safely.	0.90
SA9: The people I work with support me to complete my task in a safe manner.	0.81
SA10: The people I work with share safety rules and instructions with me.	0.79
SA11: The level of safety cooperation between the people I work with is satisfactory.	0.73
SA12: I feel satisfied with the attention given to safety in any training I have had.	0.86
SA13: I learned more in any safety training I have had.	0.83
SA14: I am satisfied with the adequacy of the level of training I have had.	0.94
SA15: Overall, I think I work safely.	0.96
SA16: I think I comply with the workplace safety rules and instructions.	0.80
SA17: The people I work with are satisfied with the information they get about safe working.	0.73
SA18: The people I work with are satisfied with the safety inspection information.	0.84
SA19: The people I work with are satisfied with the ways of presenting of safety information.	0.76
SM: Safety Motivation	
SM1: I feel that it is important to maintain safety at all times.	0.73
SM2: I believe that safety in the workplace is a very important issue.	0.85
SM3: I feel that it is necessary to make an effort to reduce accidents and incidents in the workplace.	0.80
SM4: I believe that safety that can be compromised to increase production.	0.81
SM5: I feel that it is important to encourage others to use safe practices.	0.86
SM6: I feel that it is important to promote safety programs.	0.77
SK: Safety Knowledge	0.72
SK1: I know how to perform my job in a safe manner.	
SK2: I know how to use safety equipment and standard work procedures.	0.91
SK3: I know how to maintain or improve workplace health and safety.	0.88
SK4: I know how to reduce the risk of accidents and incidents in the workplace.	0.83
SK5: I know all the hazards associated with my job and the necessary precautions to be taken while doing my job.	0.96
SK6: I know what to do and when to report if a potential hazard is noticed in my workplace.	0.86
SC: Safety Compliance	
SC1: I use all necessary safety equipment to do my job.	0.83
SC2: I carry out my work in a safe manner.	0.79
SC3: I follow correct safety rules and procedures while carrying out my job.	0.86
SC4: I ensure the highest levels of safety when I carry out my job.	0.90
SC5: It is always practical to follow all safety rules and procedures while doing a job.	0.75
SP: Safety Participation	
SP1: I help my coworkers when they are working under risky or hazardous conditions.	0.93
SP2: I always point out to the management if any safety-related matters are noticed in my company.	0.84
SP3: I make an effort to improve the safety of the workplace.	0.95
SP4: I voluntarily carry out tasks or activities that help to improve workplace safety.	0.94
SP5: I encourage my coworkers to work safely.	0.87

**Table 3 ijerph-18-04196-t003:** Confirmatory factor results: reliability analysis, unidimensionality for all measures (safety leadership, individual safety attitudes safety motivation, safety knowledge, safety compliance and safety participation).

Measure Name	Item Measure No.	Cronbach’s Alpha	CFI
Safety leadership	16	0.87	0.91
Safety attitudes	24	0.81	0.95
Safety motivation	5	0.80	0.98
Safety knowledge	6	0.86	0.92
Safety compliance	4	0.92	0.90
Safety participation	6	0.88	0.94

**Table 4 ijerph-18-04196-t004:** Mean, standard deviations and correlations between output measures.

Output Measures	Mean	S.D.	1	2	3	4	5	6	7	8	9
1. Age ^a^	3.01	0.82	1								
2. Work experience ^b^	3.21	0.43	0.73 ***	1							
3. Education degree ^c^	3.83	0.56	0.35 **	0.46 ***	1						
4. SL	4.04	0.52	0.12	0.24*	0.43 ***	1					
5. SA	4.28	0.68	0.46 ***	0.28 **	0.28 **	0.49 ***	1				
6. SM	4.13	0.44	−0.15	−0.09	−0.25 *	0.56 ***	0.66 ***	1			
7. SK	4.01	0.53	0.18	0.31 **	0.39 **	0.40 ***	0.44 ***	0.53 ***	1		
8. SC	4.38	0.51	−0.06	0.26 *	0.45 ***	0.62 ***	0.34 **	0.48 ***	0.50 ***	1	
9. SP	4.18	0.61	−0.16	−0.36 *	0.33 **	0.38 **	0.51 ***	0.29 **	0.35 **	0.70 ***	1

^a^ Age score based on 1 = less than or equal to 20 years, 2 = 21–30 years, 3 = 31–40 years, 4 = 41–50 years and 5 = more than 50 years; ^b^ Work experience score based on 1 = less than 3 years, 2 = 3–5 years, 3 = 6–10 years, 4 = 11–15 years and 5 = more than 15 years; ^c^ Education degree score based on 1 = Under primary school, 2 = Primary school, 3 = High school, 4 = Diploma and 5 = Degree or higher; * Significance level < 0.05; ** Significance level < 0.01; *** Significance level < 0.001.

**Table 5 ijerph-18-04196-t005:** Accident type statistics of the nine contractors in electrical construction projects in Saudi Arabia.

Contractor	Accident Type (2016–2019)			Total
	Minor	Major	Fatal	
**1**	190	33	3	226
2	210	28	3	241
3	308	96	2	406
4	296	123	4	423
5	135	31	2	168
6	263	72	2	337
7	321	89	4	414
8	172	101	3	276
9	314	115	4	433
Total	2209	688	27	2924

**Table 6 ijerph-18-04196-t006:** Competing structural equation models for predicting safety compliance and safety participation via safety leadership (group-level factor) and safety attitudes (individual-level factor) mediated by safety motivation and safety knowledge.

Model Condition	χ^2^	χ^2^/*df*	RMSEA	Δ χ^2^	CFI	TLI	IFI	Δ *df*
Main model: indirect main effect of SL, SA and SL × SA	673.45	1.23	0.048	-	0.962	0.952	0.963	-
Model 1: indirect main effect of SL, SA, SL × SA and direct effect of SL upon SC and SP	735.33	1.64	0.051	61.88 **	0.942	0.934	0.942	8
Model 2: indirect main effect of SL, SA, SL × SA and direct effect of SA upon SC and SP	708.42	1.41	0.050	34.97 *	0.952	0.943	0.954	4
Model 3: indirect main effect of SL, SA, SL × SA and direct effect of SL × SA upon SC and SP	678.83	1.28	0.049	5.38	0.958	0.950	0.958	2
Model 4: Indirect main effect of SL, SA and no effect of SL × SA	755.08	2.12	0.058	81.63 ***	0.924	0.917	0.925	12
Model 5: Indirect main effect of SL, SA, direct effect of SL upon SC and SP and no effect of SL × SA	744.56	1.87	0.053	71.11 ***	0.931	0.924	0.933	10
Model 6: Indirect main effect of SL, SA, direct effect of SA upon SC and SP and no effect of SL × SA	731.48	1.61	0.051	58.03 **	0.947	0.939	0.949	6

* Significance level < 0.05; ** Significance level < 0.01; *** Significance level < 0.001.

## Data Availability

All questionnaires that were observed are stored in the author’s office.

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
