# Peer review of "Safety Leadership, Safety Attitudes, Safety Knowledge and Motivation toward Safety-Related Behaviors in Electrical Substation Construction Projects"

_ijerph, 2021, doi:10.3390/ijerph18084196_

Round 1
Reviewer 1 Report
I have carefully read the manuscript titled “ Safety Leadership, Safety Attitudes, Safety Knowledge and Motivation toward Safety-Related Behaviors in Electrical Sub-station Construction Projects”. The manuscript faces an interesting topic in the occupational safety scenarios in the electrical construction field.
Some comments from my side:
In the abstract section there are a lot of concepts repeated several times. This should be revised.
Although the introduction section addresses interesting topics in the overall field of occupational accidents, it sounds quite long and the risk is to lose the attention of the reader, also the paragraphs “Safety leadership” and “Safety attitudes…” may all be summarized and included in the introduction section.
The limitations of the study should include also the lack of data on specific occupational tasks performed by the enrolled workers although engaged in the same sector, on the length of employment, as well as on different types of contractual engagement, job satisfaction, that may explain possible differences in safety leadership and safety attitudes appreciation.
From a practical perspective, which preventive strategies could be improved by these findinfgs? The author should provide some suggestion son the measures that may enhance the safety approach in this occupational sector. These may include also information plans.
Author Response
Response to Reviewer 1 Comments
Point 1: In the abstract section there are a lot of concepts repeated several times. This should be revised. 

Response 1: In the Abstract, the repetitions have been removed and made minor changes (lines 10–12 and 21–25). For line 19, instead of “… leadership and attitude and their interaction...” we used “these factors.”
Point 2: Although the introduction section addresses interesting topics in the overall field of occupational accidents, it sounds quite long and the risk is to lose the attention of the reader, also the paragraphs “Safety leadership” and “Safety attitudes…” may all be summarized and included in the introduction section.
Response 2: The Introduction is summarized: PAGE 2; lines 81–84, 89–91, and PAGE 3; 99–100, 102–105, 123–129, 140–143, and 148–151 have been removed. In addition, Safety Leadership is summarized: PAGE 4; lines 170–174 have been removed. In Safety Attitudes, Safety Motivation and Safety Knowledge, PAGE 4; lines 202–204 have been removed. Both sections have been merged with the Introduction.
Point 3: The limitations of the study should include also the lack of data on specific occupational tasks performed by the enrolled workers although engaged in the same sector, on the length of employment, as well as on different types of contractual engagement, job satisfaction, that may explain possible differences in safety leadership and safety attitudes appreciation.
Response 3: Limitation points have been added to PAGE 14, Limitations of the Study and Future Research; line 545 (“Additionally, there was a lack of objective...”) and line 552 (“Finally, factors such as a contractor’s experience in the same type of electrical construction projects…”).
Point 4: From a practical perspective, which preventive strategies could be improved by these findings? The author should provide some suggestion son the measures that may enhance the safety approach in this occupational sector. These may include also information plans.
Response 4: Proposed preventive strategies have been added to PAGE 14, Implication of the Study Findings on line 533 (“Additionally, understanding the importance of how safety leadership …”).
Minor spelling and grammar checks were also applied.

Reviewer 2 Report
The paper aims to determine the interaction between a group-level factor (i.e., safety leadership) and individual safety attitudes on individual safety knowledge, skills and motivation as factors of safety performance. The subject matter appears current and consistent with the goals of the journal. Nevertheless, some revisions to the work are deemed necessary. First of all, the article presents a limited review of the literature and several statements are not supported by references. For example:
- PAG 2: “Numerous studies (what are these studies?) have considered the effect of organizational-level safety factors (e.g., safety commitment) and individual-level safety factors (e.g., safety behaviors) in various parts of the world, but in Saudi Arabia, safety issues have yet to receive much attention. In this situation, projects are necessary to identify the proximal factors that can lead to deteriorated safety performance and increased accidents”;
- PAG 3: “Numerous researchers (specify others which) have noted that the greatest determinants of safety-related behaviors for task performance are safety knowledge, skills and motivation [20]”;
- PAG 4: “The safety leadership factor is a group-level factor that can impact the individual level and safety performance. For in-stance, effective safety leadership behavior can positively impact safety performance through discussions of safety issues with workers and by providing valuable guidance and direction for a safe workplace”.
The references are sometimes a bit "dated" (are you sure that more recent works have not produced useful contributions to the analysis?). Therefore, it is appropriate to focus the literature review more on the identification of the gap in the literature, also integrating the analysis with more recent contributions.
Figure 1 representing the structural equation modeling shows no source, so it is unclear if it is an elaboration of the authors or was reproduced from other studies.
Finally, with reference to the results presented, we invite the authors to discuss the evidence in light of the relevant literature
Author Response
Response to Reviewer 2 Comments
Point 1: PAG 2: “Numerous studies (what are these studies?) have considered the effect of organizational-level safety factors (e.g., safety commitment) and individual-level safety factors (e.g., safety behaviors) in various parts of the world, but in Saudi Arabia, safety issues have yet to receive much attention. In this situation, projects are necessary to identify the proximal factors that can lead to deteriorated safety performance and increased accidents”.

Response 1: The new recent references added to the " Numerous studies have considered the effect of … [9,10]" (line 55). Also, references added " Researchers of workplace safety have created and tested… [2,12,13]" (line 62). PAGE 2
Point 2: “Numerous researchers (specify others which) have noted that the greatest determinants of safety-related behaviors for task performance are safety knowledge, skills and motivation [20]”;.
Response 2: The new recent references added to the " Numerous researchers have noted that the greatest determinants … [20,21,22]." (line 113). PAGE 3
Point 3: PAG 4: “The safety leadership factor is a group-level factor that can impact the individual level and safety performance. For in-stance, effective safety leadership behavior can positively impact safety performance through discussions of safety issues with workers and by providing valuable guidance and direction for a safe workplace”.
Response 3: The new recent references added to the " The safety leadership factor is a group-level factor … [24]." (line 130). Also, references added to " For in-stance, effective safety leadership behavior can positively impact … [24,30]. (line 133). PAGE 4
Point 4: The references are sometimes a bit "dated" (are you sure that more recent works have not produced useful contributions to the analysis?). Therefore, it is appropriate to focus the literature review more on the identification of the gap in the literature, also integrating the analysis with more recent contributions.
Response 4: It was focused more on the literature review and added more recent references as showed in "References" section (References # 9,10,12,13,27,33 and 39) in the revised manuscript.
Point 5: Figure 1 representing the structural equation modeling shows no source, so it is unclear if it is an elaboration of the authors or was reproduced from other studies.
Response 5: PAGE 5; illustrated in page 4 line 176 " …modelling, the structural model was built in the present study…".
Point 6: Finally, with reference to the results presented, we invite the authors to discuss the evidence in light of the relevant literature.
Response 6: added more discussion in PAGE 13 (Discussion) section lines 450, 456 and 463 " …modelling, the structural model was built in the present study…".
Minor spell and grammar check applied also for the paper.
